# Effects of Pro-Inflammatory Cytokines on Hepatic Metabolism in Primary Human Hepatocytes

**DOI:** 10.3390/ijms232314880

**Published:** 2022-11-28

**Authors:** Roberto Gramignoli, Aarati R. Ranade, Raman Venkataramanan, Stephen C. Strom

**Affiliations:** 1Department of Laboratory Medicine, Division of Pathology, Karolinska Institutet, 171 77 Stockholm, Sweden; 2Department of Pathology and Cancer Diagnostic, Karolinska University Hospital, 141 83 Stockholm, Sweden; 3Cell-Based Assay Division, ScitoVation, Durham, NC 27709, USA; 4Department of Pharmaceutical Sciences, University of Pittsburgh, Pittsburgh, PA 15261, USA

**Keywords:** hepatocyte transplantation, inflammatory cytokines, hepatic functions

## Abstract

Three decades of hepatocyte transplantation have confirmed such a cell-based approach as an adjunct or alternative treatment to solid organ transplantation. Donor cell survival and engraftment were indirectly measured by hepatospecific secretive or released metabolites, such as ammonia metabolism in urea cycle defects. In cases of sepsis or viral infection, ammonia levels can significantly and abruptly increase in these recipients, erroneously implying rejection. Pro-inflammatory cytokines associated with viral or bacterial infections are known to affect many liver functions, including drug-metabolizing enzymes and hepatic transport activities. We examined the influence of pro-inflammatory cytokines in primary human hepatocytes, isolated from both normal donors or patients with metabolic liver diseases. Different measures of hepatocyte functions, including ammonia metabolism and phase 1–3 metabolism, were performed. All the hepatic functions were profoundly and significantly suppressed after exposure to concentrations of from 0.1 to 10 ng/mL of different inflammatory cytokines, alone and in combination. Our data indicate that, like phase I metabolism, suppression of phase II/III and ammonia metabolism occurs in hepatocytes exposed to pro-inflammatory cytokines in the absence of cell death. Such inflammatory events do not necessarily indicate a rejection response or loss of the cell graft, and these systemic inflammatory signals should be carefully considered when the immunosuppressant regiment is reduced or relieved in a hepatocyte transplantation recipient in response to such alleged rejection.

## 1. Introduction

There are a large number of patients awaiting liver transplantation since the availability of donor organs is limited. Thirty years of clinical infusion have led to hepatocyte transplantation (HepTx) being accepted as an alternative promising therapy for a variety of indications [1]. The most common indications of HepTx are liver-based inborn metabolic disorders [1]. These diseases are characterized by the lack of one particular enzyme or protein, giving rise to hepatic and/or extrahepatic disease. As the liver has a high redundancy in function, replacement of 5–20% of the liver with healthy donor hepatocytes can correct a wide range of inherited metabolic liver diseases, and the replacement of the whole liver by orthotopic liver transplantation may not be necessary [2].

Patients with urea cycle defects and Crigler-Najjar type 1 disorder are the most common recipients of this alternative treatment, as many clinical experiences up to now may prove [1,2]. As it is extremely difficult to track the cells and monitor their engraftment and/or proliferation inside the recipient’s liver once injected, the continued function of donor cells is inferred from the ammonia levels or conjugation activity of recipients deficient in such metabolites. Nevertheless, in cases of sepsis or viral infection, ammonia levels can significantly and abruptly increase in patients with urea cycle defects. If this occurs in a HepTx recipient, one might incorrectly suspect that the cell graft has been rejected.

Pro-inflammatory cytokines associated with viral or bacterial infections such as tumor necrosis factor-alpha (TNFα), interleukin-6 (IL6), or interleukin-1beta (IL1β) are known to affect many liver functions or synthetic activities. A number of studies suggest that inflammatory responses involving TNFα, IL6, and IL1β are involved in the downregulation of phase I enzymes (CYP) enzymes and hepatic drug transporters [3,4,5,6,7,8,9,10,11,12,13]. The regulation of CYPs in models of infection and inflammatory diseases has been studied extensively [3,4,5,6,7,8,9,10,11]. The downregulation of CYP activities or protein levels during inflammation is also generally accompanied by a decrease in the respective CYP mRNA expression [5,6]. Thus, it has been suggested that transcriptional suppression could be the primary mechanism for the decrease in CYP mRNA expression during inflammation. The downregulation of transcription factors such as hepatocyte nuclear factors, NF-κB, along with several nuclear receptors such as pregnane X receptor (PXR) and constitutive androgen receptor (CAR), have been proposed to be responsible for the suppression of CYPs by inflammatory stimuli [4,14,15]. Furthermore, PXR and CAR also regulate phase II enzymes. Hepatic phase-II-metabolizing enzymes, such as UDP- glucuronosyltransferase (UGT), play an important role in drug disposition and clearance, along with CYPs and various drug transporters. Many drugs used in transplanted patients, such as mycophenolic acid, acetaminophen, and morphine are mainly metabolized by glucuronidation. In addition, several endogenous compounds such as bilirubin, estradiol, and testosterone also undergo glucuronidation. Notwithstanding this, there is relatively little information about the regulation of phase II enzymes during inflammation. 

Most of the studies involving the use of cytokines are carried out in animals (mice, rats, or pigs) or using in vitro systems such as human intestinal or hepatic cell lines. In vitro and in vivo studies in rats indicate that pro-inflammatory cytokines decrease the uptake and secretion of bile components by altering the expression and activity of hepatic transporters, such as bile salt express pump (BSEP), sodium-dependent taurocholate transporting protein (NTCP), and multidrug-resistant-associated protein 2 (MRP2), thus leading to cholestatic conditions [16,17,18,19]. Human hepatocytes in culture contain all the necessary cofactors as well as the regulatory elements required to maintain and modulate the transporter proteins, and are hence extremely well-suited to conduct these studies.

The present work was undertaken to explore the influence of increasing concentrations of pro-inflammatory cytokines on different hepatic activities, from CYP enzymes to UGT1A1, to BSEP and ammonia metabolism using primary human hepatocytes. We examined the influence of such pro-inflammatory cytokines in primary human hepatocytes, isolated from both normal donors and patients with metabolic liver diseases. The proposed study has significance for patients following HepTx procedures in that abrupt spike in ammonia levels or reduced conjugation capacity concurrent with bacterial or viral infections. Such inflammatory events may not necessarily indicate a rejection response or loss of the cell graft, and should be carefully considered when the immunosuppressant regiment is reduced or relieved in a HepTx recipient in response to such alleged rejection. Alterations in pro-inflammatory cytokines in diseased states, such as infection, cancer, or rejection after cellular or organ transplantation, may modulate the activity of hepatic functions. Accordingly, relief of inflammation by cytokine-targeted drug therapy might have the potential to up-regulate hepatic activities and thereby increase the effects of cell-based therapy and clearance of co-administered drugs.

## 2. Results

Hepatocytes from a total of 27 human liver donors were used to conduct the experiments described in this study. Their relevant demographics and viability are listed in Appendix A. Individual donors included different forms of benign diseases (1 adenoma, 1 focal nodular hyperplasia) or malignant diseases (1 hepatocellular carcinoma, 6 colorectal metastasis, 2 unidentified forms of neoplasia), and non-diseased tissue was used. Ten (10) samples were acquired from organ donors and six from patients receiving liver transplantation for inborn errors of intermediary metabolism (1 Primary Hyperoxaluria, 2 Maple Syrup Urine Disease), including UCDs (1 OTC and 2 CPS-1 deficiencies). The age of the donors ranged from 6 months to 79 years, with mean viability equal to 77 ± 3% (range 48–98%) (Appendix A).

Cytotoxicity, as an MTT measurement, total double-strand DNA amount, and apoptosis evaluation, as Caspase 3/7 activity measurements, were performed in plated hepatocytes after exposure, alone and in combination, to the three cytokines (Figure 1).

Chronic exposure (72 h) of human hepatocytes ranging from 0.1 to 50 ng/mL resulted in significant cellular toxicity above 20 ng/mL, as compared to the untreated cells (Figure 1A). All three inflammatory cytokines have similar cytotoxic effects when the total amount of double-strain DNA or apoptotic cells is quantified in a dose escalation study (after 24 h, Figure 1B,D). Similarly, no significant differences were measured in terms of the total amount of dsDNA and apoptosis at cytokine concentrations below 10 ng/mL, even after prolonged exposure (72 h; Figure 1C,E). Considering these evaluations, the hepatic activity experiments were carried out using cytokine concentrations at or below 10 ng/mL. With all cytokine treatments, the hepatocytes showed no change in adherence characteristics and morphology by light microscopy (Figure 2).

### 2.1. Inflammatory Cytokines on Ammonia Metabolism 

Collagen-plated human hepatocytes (n = 15), including UCD preparations, were used to study ammonia metabolism. Cells were exposed to an ammonia-containing medium, and clearance was determined (Figure 3).

To determine the effects of increasing doses of inflammatory cytokines, we evaluated the ammonia metabolism in the same hepatocyte preparations in culture after three days of exposure to TNFα, IL-6, and IL-1β, at increasing inflammatory signals (0.1–10 ng/mL; Figure 3, panels A–C). A similar dose–response relationship was seen with all the studied inflammatory cytokines, with IL-1β showing the greatest amount of inhibition (49 ± 33% decrease compared to untreated), even at the lowest dose (0.1 ng/mL) (Figure 3C). 

All three inflammatory cytokines showed a dose–response relationship with the highest concentration (10 ng/mL), reducing ammonia metabolism to >50% of baseline levels after 3 days. These findings were highly significant in all cases (*p* < 0.0001). Many of the inflammatory cytokines are known to act synergistically upon their targets. However, in the studied samples, combinations of inflammatory cytokines at the highest inhibitory dose (10 ng/mL) did not demonstrate significantly stronger synergy compared to a single cytokine (Figure 3D). To chart the timing of the inhibition of ammonia metabolism, healthy human hepatocytes were treated with the highest cytokine dose (10 ng/mL) and followed over three days (Figure 3E–G). The hepatocytes showed an initial increase in ammonia metabolism on day 1 above control (44 ± 3%, 31 ± 10%, and 59 ± 10% in TNFα, IL-6, and IL-1β, respectively); followed by inhibition consistent with the dose–response studies seen on day 3 (decrease equal to 58 ± 8%, 23 ± 8%, and 42 ± 1% compared to untreated control in TNFα, IL-6, and IL-1β, respectively). 

### 2.2. Inflammatory Cytokines on CYP Activities 

Twelve (12) human hepatocytes preparations, including UCDs, were used to measure the effects of pro-inflammatory cytokines on 4 different CYP enzymes: CYP1A2, CYP2C9, CYP3A4, and CYP3A7. Since, in cells obtained from patients younger than 3 years in age (n = 5), the activity of the fetal 3A7 is usually prominent, we measured 3A4 and 3A7 enzymes using two different 3A-specific compounds (Luciferin-PFBE and Luciferin-IPA). 3A4 activity was present, at a different level, in all the preparations; while 3A7 activity was only detectable in the four preparations characterized by < 3 years in age (primary hyperoxaluria and UCDs). 

In contrast with what was observed in ammonia metabolism, where the effects of pro-inflammatory cytokines were not detected since day 2, all the preparations showed a significant decrease in CYP activities immediately after a few hours of exposure (24 h) to any pro-inflammatory cytokines (Figure 4).

A similar dose–response relationship in phase I enzymes was observed in all the cytokine treatments, with a significant inhibition even at the lowest dose (0.1 ng/mL) (Figure 5). All the cytokine combinations produced a significant suppression in CYP activities (Appendix A).

### 2.3. Inflammatory Cytokines on Phase II Enzyme (UGT1A1) Activity 

Hepatocytes (n = 8) in the 3D culture system were treated with four different concentrations of TNFα, IL-6, and IL-1β (0.3, 1.0, 5.0, and 10 ng/mL) for 72 h, and the effect on UGT1A1 activity was determined. Figure 6A shows that increasing concentrations of TNFα significantly inhibited the formation of E-3-G, with a max effect at 10 ng/mL (decrease in activity equal to 77 ± 4%, compared to control value). E-3-G formation rate was decreased to 88 ± 5%, 82 ± 5% and 73 ± 4% of control value, at 1 ng/mL, 5 ng/mL and 10 ng/mL of IL-6 (Figure 6B). Even more evident was the effect of IL-1β: increasing concentrations were reflected in the decreased formation of E-3-G up to 41 ± 3% of control, at 10 ng/mL (Figure 6C).

This effect was confirmed in five human hepatocyte preparations tested for resorufin conjugation. A significant suppressive effect in UGT1A activity was measured after exposing the cells to TNFα and IL-1β effect (Figure 6, panels D and F). Notably, a non-significant decrease was measured in IL6-treated cells (Figure 6E).

### 2.4. Inflammatory Cytokines on Drug Transporter (BSEP) Activity 

To determine the effect of culture times on expression and activity of BSEP, hepatocytes coated with Matrigel^TM^ were exposed to each cytokine (TNFAα, IL-6, and IL-1Ββ) at a concentration of 5 ng/mL for 24–72 h. [3H] taurocholate TC ([3H]TC) efflux was used as a measure of BSEP activity. Figure 6 shows the effect of the three cytokines on BSEP activity. Maximal suppression of BSEP-mediated [3H]TC efflux was observed 72 h after treating the cells with 5 ng/mL of individual cytokines. BSEP-mediated [3H]TC efflux was suppressed to 38 ± 1%, 17 ± 6% and 79 ± 4% of control value, after TNFα, IL-6, and IL-1β treatments, respectively (Figure 7, panels A–C).

From the previous results, it was found that maximum suppression of BSEP activity was observed after 72 h of cytokine treatment. Hence, an exposure time of 72 h was chosen to study the effect of cytokine concentrations on BSEP expression and activity. Hepatocytes were treated with four different concentrations of the three single cytokines (0.3, 1.0, 5.0, and 10 ng/mL) for 72 h. Cyclosporine (CsA) (10 μM), which is an inhibitor of BSEP, was used as a positive control in this study, producing a reduced [3H]TC efflux equal to 10 ± 7% of the control value. The exposure of hepatocytes to TNFα significantly decreased the [3H]TC efflux of BSEP to 5.1 ± 6.3% of untreated cells, at a concentration of 10 ng/mL (Figure 7D). The exposure to IL-6 produced a minimal decrease in BSEP mediated [3H]TC efflux, with maximal effect at 5 ng/mL (70 ± 2%) (Figure 7E). After IL-1Β β treatment, [3H]TC efflux decreased to 47 ± 16% and 27 ± 4% at concentrations of 0.3 and 10 ng/mL, respectively (Figure 7F).

## 3. Discussion

Inflammatory mediators have been shown to reduce hepatocyte metabolism and activities. In the early stages of an inflammatory response, cytokines (mainly TNFα, IL6, and IL1β) were abruptly produced by monocytes/macrophages and endothelial cells, and released into the systemic circulation, initiating the so-called acute-phase response [20,21]. Hepatocytes are influenced by these inflammatory cytokines, and we hypothesize that activation of the immune system that occurs with infection might influence the outcome of HepTx. An alteration/reduction in drug metabolism or enzymatic activity as a result of inflammation or infection may have major implications in determining the outcome of cell-based therapy, such as liver cell transplantation. A typical HepTx protocol requires an infusion of billions of hepatocytes (the total amount of hepatocytes injected in a HepTx event is calculated per body weight and, in an adult patient, may reach up to 10^10^ cells) [1,22,23,24] reflecting a potential portal thrombosis, kept under control by multiple hepatocyte infusions distributed over 24–36 h. Moreover, current clinical protocols include pre-treatment of the recipient with local radiation [23] or partial liver resection [24], and may induce an inflammatory response in the recipient. The traditional serological markers used to determine allo-rejection, such as circulating transaminases, are poorly informative due to the paucity of engrafted donor hepatocytes. Donor hepatocytes can potentially be visualized by immunohistochemistry or in situ hybridization, or indirectly by molecular techniques [25]. However, once again, the distribution of few implanted cells into host parenchyma severely limits sensitiveness, and the risks associated with such an invasive collection of bioptic material discourage similar approaches. Another indirect approach to tracking donor hepatocytes after transplantation replies on the hepato-specific function of donor cells. In inborn errors of metabolism, proficient cells offer all the missing enzymes, resulting in metabolites and synthetic molecules being released in the circulatory system and easily being detected by hematological analysis [23,24]. Inflammation arising from transient thrombotic events generated by hepatocyte infusion through a portal vein, as well as viral or bacterial infections, are known to affect liver functions, although temporarily. Drug-metabolizing enzymes and hepatic transport activities have been reported to be temporarily inferred by inflammatory mediators and may result in altered serological levels of metabolites used to indirectly evaluate donor cell status.

To investigate this possibility, we examined the effects of pro-inflammatory cytokines such as TNFα, IL-6, and IL-1β on hepatocyte functions. We focused our attention only on these three pro-inflammatory cytokines because their primary role in human hepatic metabolism has been known since 1993 [26]. 

A non-cytotoxic level of the increasing amount of these three pro-inflammatory cytokines, in combination or alone, has been validated by MTT measurement, total dsDNA amount quantification, and apoptosis evaluation in donor hepatocytes. Subsequently, physiological, and sovra-physiological levels of inflammatory cytokines have been used to determine the acute phase on human hepatocytes and related inhibitory effects on several hepatospecific functions.

Drug-metabolizing CYP activities show inter-individual variability [27]. A component of this variation is explained by the existence of polymorphic CYP genes [28]. However, certain CYPs (e.g., CYP3A4) show a wide inter-individual variability that cannot be explained by low-frequency polymorphisms alone. Other factors, such as age, hormonal status, diet, diseases, and exposure to drugs or xenobiotics, can also determine the phenotypic variability of CYPs [29,30]. Nevertheless, certain pathological states, particularly those involving inflammatory response (i.e., bacterial and viral infection), are associated with lower drug metabolism in the organism and decreased hepatic CYP content [6,7], which ultimately influences the fate and therapeutic efficacy of many drugs and, hypothetically, even cell-based therapies. Response to cytokine effects may vary between species and patients. Caution must be examined when extrapolating studies conducted on animals, and important observations in humans are critical to providing data on alternative therapies such as HepTx. In the present study, we have confirmed that the duration of pro-inflammatory cytokine exposure to human hepatocytes is an important factor affecting the downregulation of hepatic functions. The observations on CYP3A4 activity are consistent with another previous study [31], where IL-1β and IL6 were shown to downregulate testosterone metabolism in human hepatocytes. We measured similar effects in other CYP enzymes: CYP1A2, CYP2C9, and CYP3A7. Cytochrome 3A7 family member is expressed at the highest levels in fetal and early postnatal life (a situation in which the inborn error patients are characterized) [32]. Patients with diseases such as UCD, where male children born with OTC deficiencies frequently die within the first months of life without effective treatment, or other metabolic disorders severely impair normal growth or survival in early life, have been the preferred recipients of HepTx [1,22]. Such recipients, with prolonged hospitalization, have also been frequently exposed to infective risks. Thus, we extended our study by investigating pro-inflammatory cytokine effects to more hepatic functions in primary human hepatocytes. Inflammatory cytokines such as TNFα, IL-6, and IL-1β have been shown to alter metabolic enzyme levels in vitro and in vivo [33] and inhibit urea cycle function in vitro [34]. It is known that the long-term culture (up to 4 weeks) of primary hepatocytes with chronic exposure to inflammatory cytokines results in the inhibition of urea formation as well as other cellular functions [34]. However, a direct assessment of the effects of short-duration inflammatory cytokines on ammonia metabolism has never been performed. To examine the early effects of inflammatory cytokines on hepatic metabolism, we turned to the short-term culture of primary human hepatocytes derived from liver resections or transplantation from live donors, including patients affected by genetic inborn errors. We found that hepatocytes acutely exposed to TNFα, IL-6, and IL-1β result in the suppression of ammonia metabolism (Figure 2). The effect was not immediately evident after a few hours of exposure (24 h), but a significant profound suppression in ammonia clearance was observed on days 2 and 3 of inflammatory cytokine exposure, both alone and in terms of synergetic effects.

Although biotransformation capacity (phase I) is of great importance for HepTx and has been proposed as a functional criterion for selecting hepatocyte batches to address HepTx [27], little attention has been paid to conjugation activity (phase II). In humans, 16 different UGT isoforms have been classified into either 1A or 2B subfamilies [35]. Among the UGT family, UGT1A1 is involved in the glucuronidation of various endogenous substrates such as bilirubin and estradiol, and drugs such as acetaminophen, thus making it the most important UGT isoform [36,37]. Crigler-Najjar syndrome type 1 is the most severe form of this autosomal recessive condition, characterized by non-hemolytic unconjugated hyperbilirubinemia associated with the absence of hepatic UGT1A1 activity. Crigler-Najjar has been the second largest cohort of patients, recipients of HepTx, and the most successful disease corrected by HepTx [1,24]. In the literature, limited data have been published on the effects of pro-inflammatory cytokines on the expression and activity of UGT1A1, and these mostly focused on rats and pigs, limiting the analysis to UGT mRNA levels or activity only [38]. To our knowledge, only a single study, performed more than 20 years ago on human liver biopsies, showed a decrease in UGT1A4, UGT2B4, and UGT2B7 mRNA levels during inflammation [39]. 

During the years, we collected a large amount of data with the aim of better characterizing the hepatocyte suspension before infusion [27]. Our results are consistent with the report conducted by Monshouwer and colleagues in pig hepatocytes: all three pro-inflammatory cytokines showed inhibition of the estradiol-3-glucuronide formation rate and resorufin conjugation activity, thus inhibiting UGT1A1 activity to a different extent. The lowered UGT1A1 activity during the inflammation process can decrease the conjugation of bilirubin and increase the concentration of bilirubin in bile and serum. The metabolism of drugs that are conjugated by UGT1A1 (i.e., immunosuppressant) will be decreased during inflammation, requiring adjustments in dosing in patients. Still, no general consensus exists on the optimal immunosuppressant regiment in patients receiving allogeneic hepatocytes, and even less is known for pediatric recipients. Refining immunosuppression and granting long-term correction upon HepTx is, therefore, severely impaired and at risk when inflammatory mediators alter pharmacological treatments.

A third set of experiments was performed to study inflammatory response to hepatic transporter activity. The results from preliminary studies conducted in our laboratory and by others have shown that the basal expression of transporters such as BSEP, MDR1, and MRP2 is significantly higher in hepatocyte cultures coated with Matrigel^TM^ compared to normal monolayered cultures. The loss of tight junctions during the hepatocyte isolation procedure results in a loss of cellular polarity and is believed to affect the expression and activity of some drug transporters by affecting the cellular localization of transporters. The application of an extracellular 3D matrix results in a cuboidal, polar hepatocyte structure and results in the relocalization of transporter proteins such as MDR1, MRP2, and BSEP in the hepatic canalicular membrane [40,41,42,43]. 

As the efflux of taurocholate into bile canaliculi is mainly mediated by BSEP, [3H]TC was used to measure BSEP activity. After treatment with cytokines, [3H]TC efflux was measured in 3D cultures of hepatocytes in the absence and presence of cations containing calcium and magnesium. Only those cells exposed to cation-containing medium retained the tight junctions that allow for the formation of canalicular structure [44,45]. [3H]TC, in cells without cations, accounts for all the processes (passive leakage and active transport) by which [3H]TC enters the media. Efflux measured in cells exposed to a cation-containing medium reflects only the passive leakage of [3H]TC from the cells. Thus, the difference in these two values will account for the BSEP-mediated transport of [3H]TC into the canalicular spaces, which has been used as an indirect measure of BSEP activity. In the present study, we found that the duration of cytokine exposures to human hepatocytes is an important factor affecting the downregulation of BSEP activity. In the human hepatocyte system, exposure to all cytokines studied in this work showed maximum downregulation of the BSEP-mediated efflux of [3H]TC after 72 h of exposure. 

In summary, different cytokines exhibit different inhibition potentials. Alteration of these cytokines in diseased states such as infection, cancer, or rejection after transplantation, might modulate hepatic activity, impacting the fate of transplanted hepatocytes in HepTx therapy and/or the pharmacokinetics of substrate drugs.

During the years, we collected a great amount of data (on more than 100 human hepatocyte preparations), with the aim of better characterizing the hepatocyte suspension before infusion and carefully pairing the recipient’s needs [46]. In our clinical HepTx program, we routinely evaluated and analyzed cell batches before infusion [27]. We have been performing the same qualitative procedures to “validate” the cell suspension not only in phases I to III, but also in ammonia metabolism. We measured hepatic activities immediately before infusion and after long-term culture to indirectly evaluate the functions of the injected cells. Indeed, we experienced (particularly from cirrhotic or inborn error patients) a low or null activity immediately after cell isolation, followed by a high activity (recovery) after a few days in culture [32]. This effect has been monitored in many hepatocyte preparations, even in the absence of specific inducers, to prove that, in many situations, hepatocytes “meet” functional suppressing signals from the surrounding environment, resulting in a negative effect on cellular therapy outcome as HepTx. As the human liver is the most important organ involved in drug metabolism and clearance, changes in the activities of metabolic enzymes and transporters, commonly experienced in HepTx indications, have the potential to reduce clinical effects in donor cells and alter significantly immunosuppressive drug effects. Additional studies are required and recommended to clarify and elucidate the molecular pathways involved in inflammation, not necessarily in relation to the cell treatment. Pro-inflammatory cytokines released by events not associated with HepTx (such as viral or bacterial infections) should not affect or drive clinical decisions in regard to the immunosuppressive regimen and/or cell-based treatments. 

## 4. Materials and Methods

### 4.1. Primary Hepatocyte Cultures

All human liver tissues were collected following ethical and institutional guidelines (IRB protocol 0411142, University of Pittsburgh, Pittsburgh, PA, USA). The tissue dissociation and subsequent hepatocyte isolation procedures used in this study were performed as previously described [46]. Briefly, the major hepatic vessels were canulated and sutured on the cut parenchyma surface. A three-step perfusion by a peristaltic pump was performed, followed by mechanical tissue disruption and filtration to produce a cellular suspension. Using a three times low-speed centrifugation (80× *g*) we obtained a final hepatocyte suspension. Cell viability was assessed immediately after isolation by mixing an aliquot of the final cell suspension with an equal volume of 0.4% (*w*/*v*) trypan blue in phosphate-buffered saline. Hepatocytes were allowed to adhere for 2 h with Hepatocyte Maintenance Media (HMM; Lonza, Walkersville, MD, USA) containing 5% bovine calf serum (Sigma Aldrich, St. Louis, MO, USA) and antibiotics-antimycotics (BioGemini). This was followed by maintenance culture for 5 days in serum-free HMM on collagen-coated (0.03 mg/mL concentration) or Matrigel^TM^-coated (0.233 mg/mL) plates. For the purpose of this study, cells were analyzed immediately after isolation and after 2 days in culture, during which hepatocytes were exposed to TNFα, IL-6, and IL-1β (0–50 ng/mL) for 24, 48, and 72 h, in single and in combination.

### 4.2. Cytotoxicity Evaluation

Cytotoxicity was measured by MTT assay. Following cytokine exposure, media were aspirated, 10% *v/v* of 5 mg/mL 3-(4,5-dimethylthiazol-2-yl)-2,5-diphenyltetrazolium bromide (MTT) was added to HMM, and cells were incubated for 30 min. Subsequently, the medium was aspirated, and cells were washed twice with HMM. An equal volume of isopropanol was added and shaken gently for 2 min. Two hundred microliters of this solution were transferred to a 96-well plate, and the absorbance was measured at 570 nm (Synergy HT, BioTek Instruments, Winooski, VT, USA). The results are expressed as a percentage of the value in untreated cells.

### 4.3. Apoptosis Evaluation

Caspase-3 and Caspase-7 were measured with a luminescent assay, Caspase-Glo^®^ 3/7 (Promega Corporation, Madison, WI, USA) according to the manufacturer’s instructions, with minor modifications previously described [47]. Results are expressed as LCU/min and normalized to a million viable cells.

### 4.4. Ammonia Metabolism Assay

The direct colorimetric determination of ammonia was determined on cell supernatant after 2 h of incubation with 1 mM ammonium chloride (Sigma Aldrich) diluted in a culture medium supplemented with 1 mM Ornithine (Sigma Aldrich) as previously described [27]. After incubation, the culture medium was clarified, and supernatants were read immediately or frozen. For quantitative determination of ammonia, a direct colorimetric method was used, and samples read at wavelengths from 560 nm (uQuant, BioTek Instr., Inc.). Linearity was confirmed by preparing a calibration curve in every reading. Results are expressed as nmol/min and normalized to protein content.

### 4.5. Cytochrome P450 Enzyme Activities

Cell-based assays specific for specific CYP (Cyp- Glo™1A2; Cyp- Glo™2C9; Cyp- Glo™3A4; Cyp- Glo™3A4/7) were used according to the manufacturer’s instruction and with minor modifications previously described [27]. Results are expressed as Luminescent Counting Unit (LCU)/min and normalized to the double-strand DNA content measured after the Glo™-assays are complete.

### 4.6. Phase II in Vitro Evaluation

Phase II activity was determined by estradiol-3-glucuronide (E-3-G) quantification by HPLC and by the metabolism of the fluorescent compound resorufin as previously described. Briefly, UGT1A1 activity was quantified by estradiol metabolism: after cytokine treatment, cells were washed with 1.5 mL of fresh medium for 1 h and then incubated in 1.5 mL of medium containing 250 mM estradiol for an additional 60 min. After incubation, the culture medium was clarified, and supernatants, along with cell lysate, were sampled and stored at −80 °C for estradiol-3-glucuronide (E-3-G) determination by HPLC and protein determination. Results are expressed as a percentage normalized to untreated controls.

Resorufin conjugation capacity was assessed by reduction in fluorescent signal selectively performed by UGT1A enzymes in human hepatocyte culture and determined as previously described [27]. Results are expressed as a percentage of reduction normalized to fresh solution.

### 4.7. Evaluation of BSEP Activity

As efflux of taurocholate into bile canaliculi is mainly mediated by BSEP, [3H]-taurocholate ([3H]TC) was used to measure BSEP activity. Evaluation of BSEP activity was performed as previously described [45]. Briefly, at the end of the treatment with cytokines, HMM was replaced with Hank’s balanced salt solution (HBSS) containing cations (calcium and magnesium) for 20 min. After this period, 1 μM [3H]TC was added to fresh HBSS (with cations) for 20 min. Uptake was stopped by aspirating the buffer solution and cells were washed three times with ice-cold HBSS (with cations). Fresh HBSS, with and without cations, was then added to the cells for 20 min. After incubation, media was sampled and counted using a liquid scintillation counter. The difference between the two culture systems (with and without cations) accounts for the indirect measurement of BSEP-mediated transport of [3H]TC into canalicular spaces. Cells were harvested and stored at −80 °C for protein determination. Results are expressed as percentages normalized to untreated controls.

### 4.8. Protein Amount Quantification

Total cellular protein contents were measured by a commercial kit (Protein Assay kit; Bio-Rad, Richmond, CA, USA) based on Bradford’s method. This involves the addition of an acidic dye to protein solution and subsequent measurement at 595 nm with a spectrophotometer (Synergy HT). Comparison to a bovine serum albumin (Sigma-Aldrich) standard solution provides a relative measurement of protein concentration at different time points in cultured cells (24, 48, and 72 h). 

### 4.9. dsDNA Amount Quantification

Double-stranded DNA (dsDNA) quantification was performed with a Quant-iT™ PicoGreen^®^ dsDNA ultrasensitive fluorescent nucleic acid-staining kit according to the manufacturer’s instructions (Molecular Probes, Invitrogen Corporation, Camarillo, CA), as previously described [27]. After the Glo™-assays were complete, each well was incubated with an equal volume of Quant-iT™ PicoGreen^®^ in TE buffer and the fluorescence intensity was read on a fluorescent spectrometer (Synergy HT) at an excitation wavelength of 488/15 nm and an emission wavelength of 528/20 nm. dsDNA concentration was quantified by interpolating the A528 values for the unknowns from a standard curve of lambda DNA using the equation (dsDNA [mg/mL] = 0.1057 × A528—61.322; R^2^ = 0.9941).

### 4.10. Statistical Analyses

All the experiments were conducted using biological triplicates. Summary statistics were calculated for all data. Two-sided Student’s *t*-test or one-way analysis of variance (ANOVA) with a post hoc Dunnett’s procedure were used for comparison of means. *p*-values less than 0.05 were considered to indicate statistical significance. All the results are expressed as mean + standard deviation (SD) or standard error (SEM), where stated. Statistical analyses were performed using Prism 5 (GraphPad Software, La Jolla, CA) software package.

## Figures and Tables

**Figure 1 ijms-23-14880-f001:**
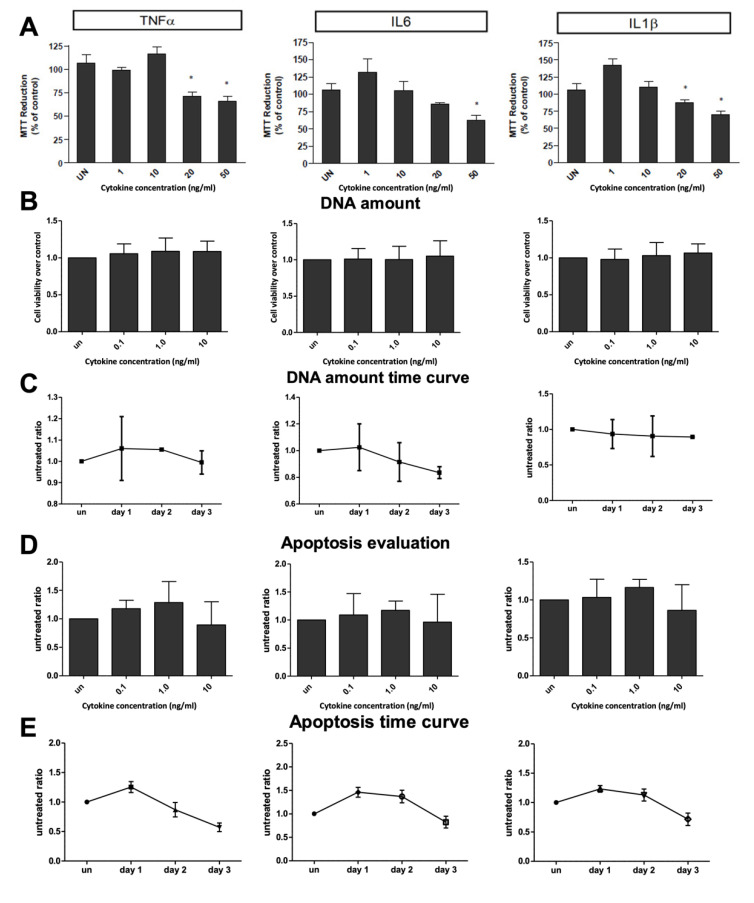
Toxicity evaluation. (**A**) MTT assay performed after 72 h cytokine exposure (0–50 ng/mL); (**B**) total double-strand DNA amount quantification after 24 h cytokine exposure (0–10 ng/mL), and (**C**) after 24, 48, 72 h cytokine exposure at higher concentration (10 ng/mL); (**D**) apoptosis evaluation measured by Caspase 3/7-Glo assay after 24 h cytokine exposure (0–10 ng/mL), and (**E**) after 24, 48, 72 h at higher concentration (10 ng/mL). * Significantly different from control (*p* < 0.05).

**Figure 2 ijms-23-14880-f002:**
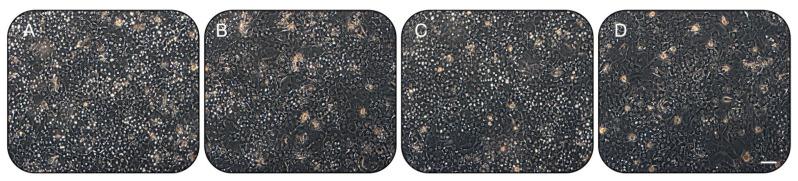
Representative phase contrast photographs for human hepatocytes cultured for 3 days in regular culture medium (**A**), or medium supplemented with (**B**) TNFα (10 ng/mL); (**C**) IL6 (10 ng/mL); (**D**) IL1β (10 ng/mL). The scale bar represents 50 µm.

**Figure 3 ijms-23-14880-f003:**
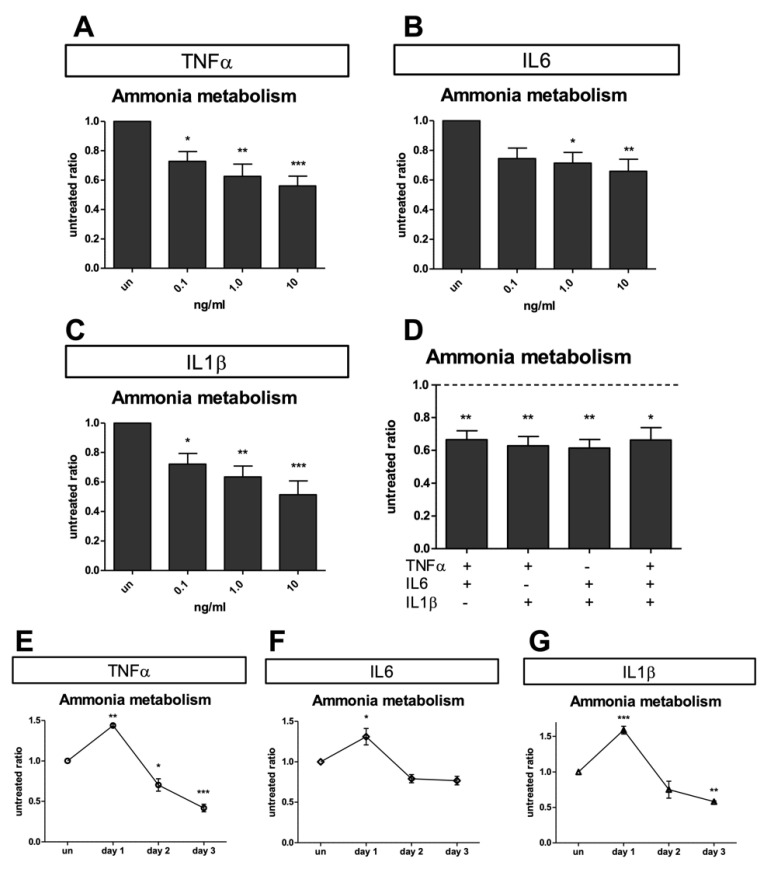
Inhibition of ammonia clearance in human hepatocytes by 72 h cytokine exposure: (**A**) TNFα (0–10 ng/mL); (**B**) IL6 (0–10 ng/mL); (**C**) IL1β (0–10 ng/mL); (**D**) combination of various inflammatory cytokines at 10 ng/mL each. Inhibition of ammonia clearance in human hepatocytes by exposure for 24, 48, and 72 h at 10 ng/mL of (**E**) TNFα, (**F**) IL-6 and (**G**) IL-1β. * *p* < 0.01; ** *p* < 0.001; *** *p* < 0.0001.

**Figure 4 ijms-23-14880-f004:**
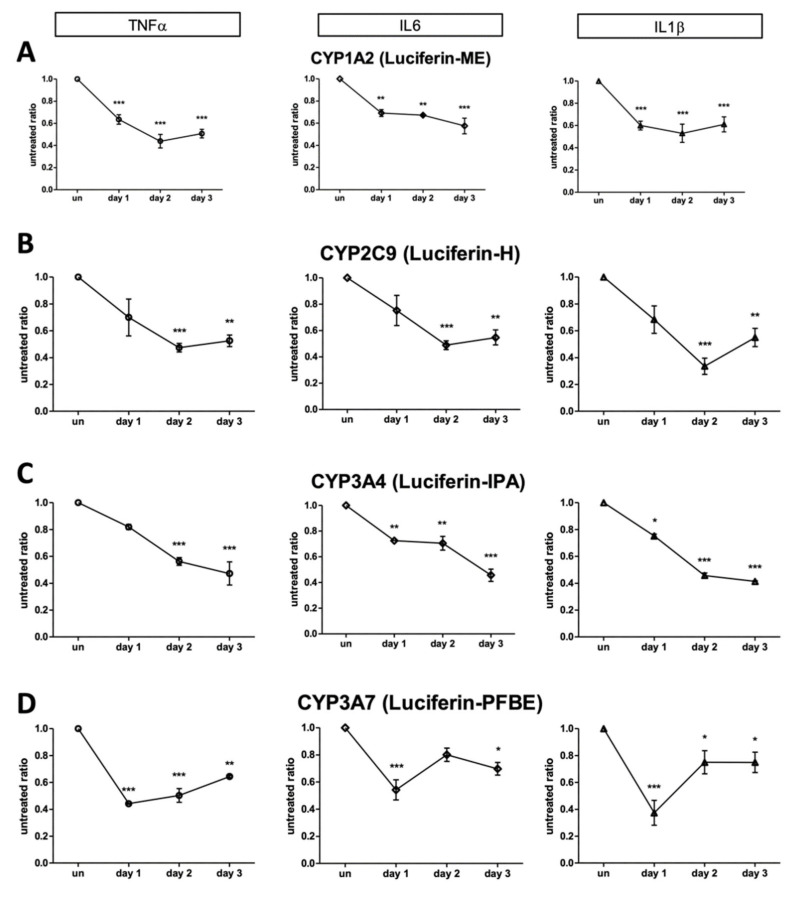
Specific CYP activity inhibitions in human hepatocytes by 24, 48, and 72 h cytokine exposure (10 ng/mL): (**A**) CYP1A2 after exposure to TNFα, IL6, IL1β; (**B**) CYP2C9 after exposure to TNFα, IL6, IL1β; (**C**) CYP3A4 after exposure to TNFα, IL6, IL1β; (**D**) CYP3A7 after exposure to TNFα, IL6, IL1β. * *p* < 0.01; ** *p* < 0.001; *** *p* < 0.0001.

**Figure 5 ijms-23-14880-f005:**
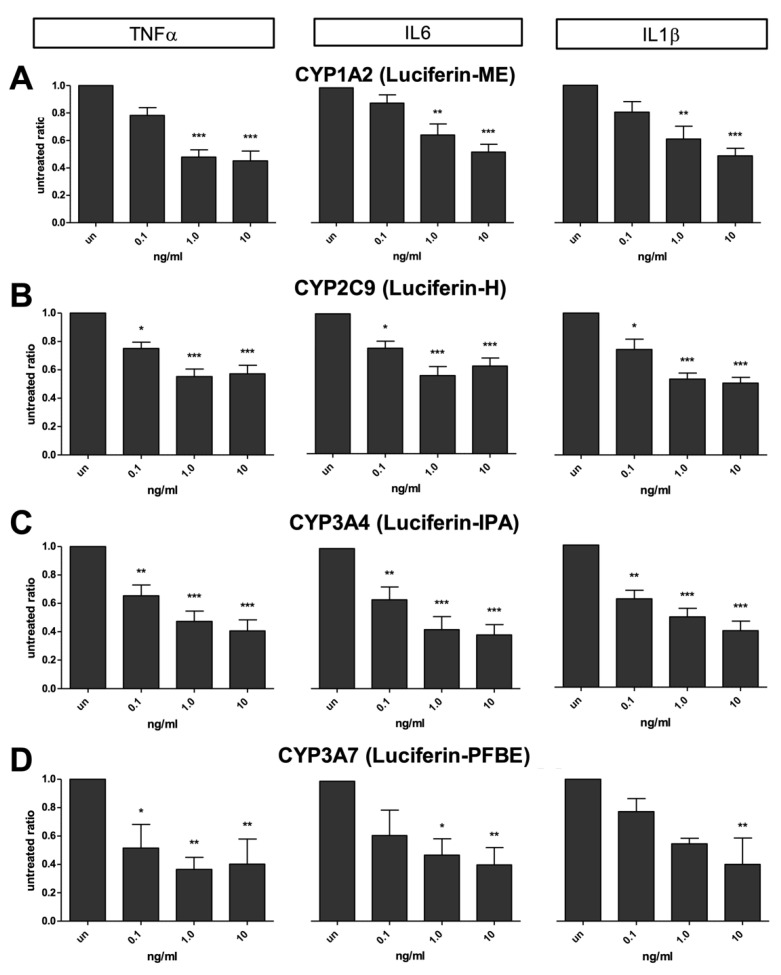
Specific CYP activity inhibitions in human hepatocytes after 72 h of exposure to different doses of pro-inflammatory cytokines (0.1, 1, and 10 ng/mL): (**A**) CYP1A2 after exposure to TNFα, IL6, IL1β; (**B**) CYP2C9 after exposure to TNFα, IL6, IL1β; (**C**) CYP3A4 after exposure to TNFα, IL6, IL1β; (**D**) CYP3A7 after exposure to TNFα, IL6, IL1β. * *p* < 0.01; ** *p* < 0.001; *** *p* < 0.0001.

**Figure 6 ijms-23-14880-f006:**
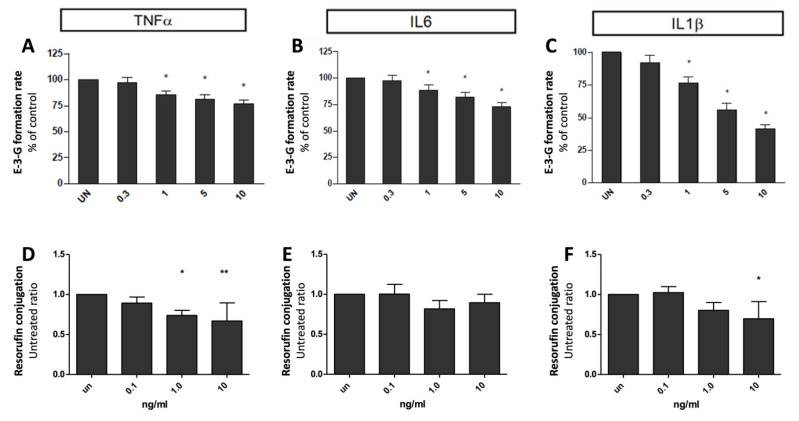
Specific UGT1A1 activity inhibitions in human hepatocytes by 72 h cytokine exposure: E-3-G formation after exposure to (**A**) TNFα (0–10 ng/mL), (**B**) IL6 (0–10 ng/mL), (**C**) IL1β (0–10 ng/mL): Resorufin conjugation after exposure to (**D**) TNFα (0–10 ng/mL), (**E**) IL6 (0–10 ng/mL), (**F**) IL1β (0–10 ng/mL). * *p* ≤ 0.05; ** *p* < 0.01.

**Figure 7 ijms-23-14880-f007:**
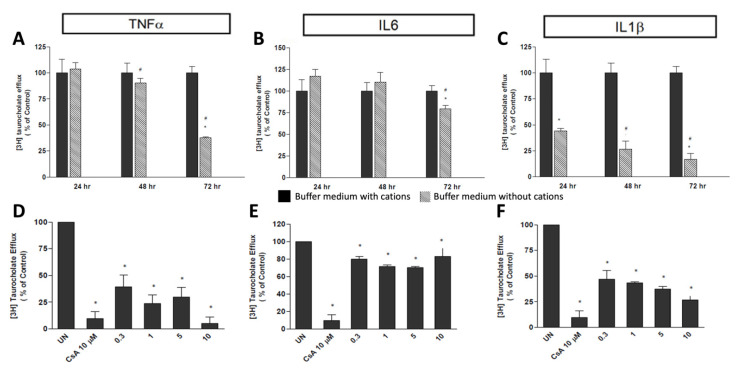
Specific BSEP activity inhibitions in human hepatocytes by 24, 48, 72 h of cytokine exposure (10 ng/mL): (**A**) TNFα, (**B**) IL6, (**C**) IL1β. Specific BSEP activity inhibitions in human hepatocytes by 72 h cytokine exposure: (**D**) TNFα (0–10 ng/mL), (**E**) IL6 (0–10 ng/mL), or (**F**) IL1β (0–10 ng/mL). * Significantly different from control (*p* < 0.05); ^#^ significantly different from 24 h value (*p* ≤ 0.05).

## Data Availability

The data presented in this study are available in this paper and Appendix A.

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
