# Peer review of "Effects of Pro-Inflammatory Cytokines on Hepatic Metabolism in Primary Human Hepatocytes"

_ijms, 2022, doi:10.3390/ijms232314880_

Round 1

Reviewer 1 Report

The work presented by Gramignoli et al. wants to investigate the effect of the increase of pro-inflammatory cytokines on hepatic activities and has been carried out on primary human hepatocytes isolated from patients with different diseases compared to primary human hepatocytes from healthy donors. The results obtained are interesting in view of understanding the possible outcome after cell therapy, in response to several inflammatory stimuli. Besides the high number of samples included in this study, which is very consistent, there are some questions to answer regarding the data obtained and some contents are missing. Moreover, going through the reading of this draft that is occasionally hard to follow, there are some points that need to be reviewed and better explained and/or simplified to not lose the point.

I suggest correcting and improve the draft according to the following points.

1)      The authors claim that hepatocytes have been isolated as “previously described”, anyway in this work is completely missing a characterization of the hepatocytes obtained: it would be valuable to add at least representative pictures of the plated cells before and after cytokines treatment, over time, for both healthy and patients, to understand the morphology of the hepatocytes and a phenotypical characterization with flow cytometry and/or immunofluorescence.

2)      The table 1 is missing, so is not possible to understand the viability that is cited in line 102.

3)      In figure 1 panel A is missing the unit of x-axis, while y-axis should be substituted with “cell viability” or “viability over control”.

4)      In figure 1, panel B the amount of DNA at the highest concentration of cytokines (10 ng/ml) should correspond to the day 3 time point in panel C, which takes into consideration only 10ng/ml. Maybe is my limitation in understanding the point but is not clear how this measurement has been done and why. The measured ds DNA is the amount released in the medium or the total DNA from hepatocytes? Is missing the n corresponding to the sample variability: are these measurements made in triplicates for each sample or not?

5)      In Figures 3, 5 and 6 are missing the letters to refer to specific panels. Figure 4 is missing or does not exist? It is cited in line 171 but not present in the manuscript. Please check the order of the figures. Supplementary fig 1 has no caption. Morphology of cells cited in line 127 should be reported in the supplemental figure1 but is missing this part.

6)      In line 162 the authors refer to the reduction of CYP activities as “in contrast to ammonia metabolism” which seems to be inhibited as well according to the previous graph in figure2. Could you explain better this sentence?

7)       In fig 6 is missing the legend which could help in clarifying the time-point graphs.

8)      From lines 211 to 215 is described the change in BSEP transporter activity according to the cytokines treatment: as shown in the graph I would not put the reader’s attention on the second one (IL-6 treatment), where the reduction is lower and not comparable with TNF-a or IL1b. In the same picture (fig.6) and or paragraph is not mentioned the number of samples or replicates performed for the analysis, which creates a no clear view of the paper.

9)      Finally, to summarize all the analyses done, with a critical view of the number included, it would be helpful for the reader to design a table where is indicated the specific number of samples used in all the types of experiments.  

Author Response

We would like to thank all the revisors for their time in reading our manuscripts and their important suggestions, aimed to improve the quality of our message and results description. We are addressing point by point each issue raised by the reviewers and pointing out where in the manuscript the modifications were made, when necessary.  The modified sections are highlighted in yellow on the resubmitted manuscript for ease of identification. 

Reviewer nr 1

The work presented by Gramignoli et al. wants to investigate the effect of the increase of pro-inflammatory cytokines on hepatic activities and has been carried out on primary human hepatocytes isolated from patients with different diseases compared to primary human hepatocytes from healthy donors. The results obtained are interesting in view of understanding the possible outcome after cell therapy, in response to several inflammatory stimuli. Besides the high number of samples included in this study, which is very consistent, there are some questions to answer regarding the data obtained and some contents are missing. Moreover, going through the reading of this draft that is occasionally hard to follow, there are some points that need to be reviewed and better explained and/or simplified to not lose the point.

We thank the Reviewer for such comments, and we confirmed we revised the manuscript to solve any issues and to smooth up the reading process.

I suggest correcting and improve the draft according to the following points.

1)      The authors claim that hepatocytes have been isolated as “previously described”, anyway in this work is completely missing a characterization of the hepatocytes obtained: it would be valuable to add at least representative pictures of the plated cells before and after cytokines treatment, over time, for both healthy and patients, to understand the morphology of the hepatocytes and a phenotypical characterization with flow cytometry and/or immunofluorescence.

Re: We would like to thank the reviewer for the useful comment. We included in the Results a new Figure (Figure 2, previously listed as Supplementary Figure 1). Figure 2 contains representative phase contrast photographs of human hepatocytes exposed to pro-inflammatory cytokines or in standard culture conditions. Similar images have been previously published by our groups, where human hepatocytes isolated from cadaveric organs or explanted livers, but also cells isolated from liver resection or pathological conditions have been isolated and seeded in culture (Gramignoli et al. Cell Transplant 2012; 21(6):1245-60; Gramignoli et al. Stem Cell Res 2013; 11(1):563-573; Jorns et al. Cell Transplant 2014; 23(8):1009-18; Gramignoli et al. Cell Transplant 2014; 23(8):1009-18). In order to minimize the number of images and Figures, we included in the current manuscript one representative image for every cytokine condition. We hope the Revisor would find sufficient and not redundant with our previous publications. 

2)      The table 1 is missing, so is not possible to understand the viability that is cited in line 102.

Re: We deeply apologize for such an oversight. Table 1 is now part of the Revised manuscript (page 3)

3)      In figure 1 panel A is missing the unit of x-axis, while y-axis should be substituted with “cell viability” or “viability over control”.

Re: Thanks to the reviewer to point out an unclear detail in our graphs. We corrected Figure 1A accordingly: FIG.1A, Y axis title is now “cell viability over control”, while X axis title is “cytokine concentration (ng/ml)”

4)      In figure 1, panel B the amount of DNA at the highest concentration of cytokines (10 ng/ml) should correspond to the day 3 time point in panel C, which takes into consideration only 10ng/ml. Maybe is my limitation in understanding the point but is not clear how this measurement has been done and why. The measured ds DNA is the amount released in the medium or the total DNA from hepatocytes? Is missing the n corresponding to the sample variability: are these measurements made in triplicates for each sample or not?

Re: We apologize for such a misunderstanding. The panel B in Figure 1 is actually reporting the values for double-strand DNA after 24 hours, to exclude a toxic effect generated by inflammatory cytokines. In previous reports, it has been described irreversible cytological damage immediately after 8-24hrs by inflammatory mediators (Ghonem N, et al. Am J Transplant. 2011;11(11):2508). Thus, we found relevant to check toxic effects in our experimental settings to exclude such detrimental effects on our primary cells. The correction has been included in the Figure legend. The assay we have implemented is actually measuring double-strand DNA in intact viable cells. The cells are lysed, and DNA quantified. All our experiments have been run in triplicate (as now stated in Materials and Methods). In panel C we are describing a minimal loss in viable cells after prolonged exposure to the highest concentration of TNFalpha or IL1beta or IL6.

5)      In Figures 3, 5 and 6 are missing the letters to refer to specific panels. Figure 4 is missing or does not exist? It is cited in line 171 but not present in the manuscript. Please check the order of the figures. Supplementary fig 1 has no caption. Morphology of cells cited in line 127 should be reported in the supplemental figure1 but is missing this part.

Re: We apologize for such an oversight. We have edited Figures 3, 5, and 6 including all the panel references. We also include an additional figure (Figure 4), previously mentioned but not included inn R0. In Figure 4, specific CYP activity inhibitions in human hepatocytes after 72 hours of exposure to different doses of pro-inflammatory cytokines (0.1, 1, and 10 ng/ml). The hepatocyte morphology is now visible in Figure 2 and briefly described in Figure Legend. Supplementary Figure 1 has been edited as suggested.

6)      In line 162 the authors refer to the reduction of CYP activities as “in contrast to ammonia metabolism” which seems to be inhibited as well according to the previous graph in figure2. Could you explain better this sentence?

Re: We apologize for such a lack of clarity. We have modified the text as follows:

In contrast with what was observed in ammonia metabolism, where the effects of pro-inflammatory cytokines were not detected since day 2, all the preparations showed a significant decrease in CYP activities immediately after a few hours of exposure (24hrs) to any pro-inflammatory cytokines (Figure 4).

7)       In fig 6 is missing the legend which could help in clarifying the time-point graphs.

Re: We thank the Reviewer for such a request which clearly improves the clarity of Figure 6. We included a legend describing bar shade.

8)      From lines 211 to 215 is described the change in BSEP transporter activity according to the cytokines treatment: as shown in the graph I would not put the reader’s attention on the second one (IL-6 treatment), where the reduction is lower and not comparable with TNF-a or IL1b. In the same picture (fig.6) and or paragraph is not mentioned the number of samples or replicates performed for the analysis, which creates a no clear view of the paper.

Re: We thank the Reviewer for the possibility to clarify and provide a better description of the experimental procedure (with additional details in Statistical analyses) and results. Dose escalation results are now described more in detail, and in the revised version, on page 8, we can read:

Exposure of hepatocytes to TNFα significantly decreased the [3H]TC efflux of BSEP to 5.1±6.3% of untreated cells, at a concentration of 10 ng/ml (Fig.6D). The exposure to IL-6 produced a minimal decrease in BSEP mediated [3H]TC efflux, with maximal effect at 5 ng/ml (70±2%)(Fig.6E). After IL-1Β β treatment, [3H]TC efflux decreased to 47±16% and 27±4% at concentrations of 0.3 and 10 ng/ml, respectively (Fig.6F).

9)      Finally, to summarize all the analyses done, with a critical view of the number included, it would be helpful for the reader to design a table where is indicated the specific number of samples used in all the types of experiments.

Re: We thank the Reviewer for a such suggestion. We included modifications in the revised version that hopefully satisfy this need

Reviewer 2 Report

The manuscript describes the effects of rejection-related cytokines on the hepatic activities of human hepatocytes that were intended for use of hepatocyte transplantation. The contents are interesting and helpful to consider the perioperative regimen of immunosuppression. Unfortunately, there are some simple syntax errors. Please refer to the attached file and revise the text.

Author Response

We would like to thank all the revisors for their time in reading our manuscripts and their important suggestions, aimed to improve the quality of our message and results description. We are addressing point by point each issue raised by the reviewers and pointing out where in the manuscript the modifications were made, when necessary.  The modified sections are highlighted in yellow on the resubmitted manuscript for ease of identification. 

Reviewer nr 2

The manuscript describes the effects of rejection-related cytokines on the hepatic activities of human hepatocytes that were intended for use of hepatocyte transplantation. The contents are interesting and helpful to consider the perioperative regimen of immunosuppression. Unfortunately, there are some simple syntax errors. Please refer to the attached file and revise the text.

We thank the Reviewer for the comments and corrections applied to the pdf file. We went through all the suggestions and requests and included such corrections in the revised manuscript. We can confirm we edited the Figure and Figure legends as requested, we fixed the capital letters in “hepatocyte transplantation”, and we converted (missing) Supplementary Figure 1 into Figure 2. We revised the last sentence in the abstract, which now sounds as: “Such inflammatory events do not necessarily indicate a rejection response or loss of the cell graft, and these systemic inflammatory signals should be carefully considered when the immunosuppressant regiment is reduced or relieved in a HepTx recipient in response to such alleged rejection”

Reviewer 3 Report

In the present study, Gramignoli et al. analyzed the influence of pro-inflammatory cytokines in primary human hepatocytes isolated  from  both  normal  donors  or  patients  with  metabolic  liver  diseases. The topic is attractive, and the findings in the present study would contribute to the advancement of clinical practice.

Unfortunately, however, although the author showed that phase II/III and ammonia metabolism was suppressed after the exposure to pro-inflammatory cytokines in primary hepatocytes, the mechanisms were not demonstrated at all. I strongly suggest that the paper should be re-submitted after some possible mechanisms such as the expressional alterations of metabolic enzymes are examined.

Author Response

We would like to thank all the revisors for their time in reading our manuscripts and their important suggestions, aimed to improve the quality of our message and results description. We are addressing point by point each issue raised by the reviewers and pointing out where in the manuscript the modifications were made, when necessary.  The modified sections are highlighted in yellow on the resubmitted manuscript for ease of identification. 

Reviewer nr 3

In the present study, Gramignoli et al. analyzed the influence of pro-inflammatory cytokines in primary human hepatocytes isolated  from  both  normal  donors  or  patients  with  metabolic  liver  diseases. The topic is attractive, and the findings in the present study would contribute to the advancement of clinical practice.

Unfortunately, however, although the author showed that phase II/III and ammonia metabolism was suppressed after exposure to pro-inflammatory cytokines in primary hepatocytes, the mechanisms were not demonstrated at all. I strongly suggest that the paper should be re-submitted after some possible mechanisms such as the expressional alterations of metabolic enzymes are examined.

We thank the Reviewer for reading our study and for the important suggestion on the mechanistic description in support of our analyses. We would respectfully like to highlight as the present study has been designed and conducted to support and guide the clinical infusion of human hepatocytes. In our clinical practice, we found quite misleading the systemic signs of inflammation when read as an indication of donor rejection or loss. The aim of our study was to draw attention to hepatocyte metabolism and synthetic activities in response to pro-inflammatory cytokines. We agree and are fully aware that the present study has not addressed any molecular pathways, nor was intended to. We aim to support and better refine the clinical practice where allogeneic hepatocytes are transplanted to support hepatic activities or provide missing enzymatic activities in patients with inborn errors of metabolism. Our study has the ambition of drawing more attention to inflammatory events, not necessarily in relation to allograft rejection or recognition, before withdrawing immunosuppression and calling the cell therapy a failure. Pro-inflammatory cytokines released by non-associated events with the cell infusion (such as viral or bacterial infections) can affect clinical judgment. We performed a study to prove as such released cytokines may significantly affect several critical liver functions, including the enzymes and transporters we intend to correct or provide by administering proficient door cells. Additional studies are undoubtedly needed to clarify the mechanistic pathways involved in such cytokine-driven effects. For such reason, we included a sentence in the Discussion, as the Reviewer suggested, highlighting such limitation. At page 11, we can now read: “Additional studies are required and recommended to clarify and elucidate molecular pathways involved in inflammation, not necessarily in relation to the cell treatment. Pro-inflammatory cytokines released by events non-associated with the hepatocyte infusion (such as viral or bacterial infections) should not affect or drive clinical judgment and decision in regard to the immunosuppressive regimen and cell-based treatments”.

Round 2

Reviewer 1 Report

"The authors answered the questions arising from the first revision to the manuscript and they corrected the draft accordingly. I don’t think the presence of representative pictures of hepatocytes (which is, anyway, not clear) is redundant with previously published works of the group, but a characterization performed with cytometry or immunofluorescence is still missing. So far, from these dark pictures is not possible to get the morphology of the cells. That characterization is missing also in previously published works.  With respect to the last suggestion I made, “design a table to indicate the number/type of samples used in all the experiments”, the authors only added the number of samples in 2 sections of the draft. My further question is related to the analysis made on all the hepatocytes isolated: all the samples were pulled and analyzed together. I can understand that availability of donors is a limiting factor for the project, but at the same time using hepatocytes from patients that have developed cancer (in a different body compartment) it could be risky for future prospective of cell transplantation thus not feasible. My suggestion of indicating where (in which experiments/analysis) the specific samples were used, was directed to understand whether a pre-existent pathology could interfere with the investigation and results obtained. Is there any difference between healthy donors and pathologic donors? Are the healthy donor hepatocytes used in a specific experiment or across all? This aspect should be discussed or at least mentioned in the results/discussion section.

Author Response

We would like to thank Revisor for the extra work taken to read again our manuscript and for all the proposed changes.

In particular, we appreciate the opportunity to clarify one important step. The revisor proposed flow cytometric or immunocytostaining as important information missing in our study. We agree with revisor that the presence of static cell markers has been largely recognized as instrumental to validate the identity of several human cell products. High-speed fluorescence-activated cell sorting (FACS) or immunocytostaining on isolated liver cells has been attempted, and our group largely reported such characterization. In our previous work, we highlighted as enzymatic digestion using collagenase frequently erases several surface proteins, jeopardizing cell identification performed by specific antibodies against surface proteins. In addition, the characteristic flow stress generated by high-speed FACS severely affects large cells such as human hepatocytes. We have reported as fluidic stress and current FACS settings (including narrow nozzle in common machines) severely limit the detection and discrimination of single events and cells generated by epithelial cells. Freshly isolated human hepatocytes have been poorly characterized by FACS so far, limiting such discrimination on immortalized lines and fetal liver cells (characterized by smaller dimensions and mononuclear status). Our group was the first one to isolate and transplant human hepatocytes in patients. Such limits forced us to opt for immunocytostaning on isolated cells. During the past 20 years, we performed more than 2,000 human hepatocyte isolations and in all cases, we got the same results: final cell suspension is >95-98% in hepatocytes. We have reported such characterization before (we are including also a representative picture for the revisor’s evaluation, previously included in another report generated by our group; please see the attachment). However, we would like to highlight as there are currently no optimal markers or antibodies, nor there is any consensus on specific and selective surface markers that can be used to clearly identify “good quality” primary human hepatocytes generated after enzymatic isolation. As we largely described and supported, metabolic and synthetic functions, characteristic of adult and functional hepatocytes, are nowadays still the best and most accurate proof of origin for liver cells. Any other cells can efficiently and specifically metabolize selective compounds characteristically processed by liver CYPs or urea cycle metabolism (compounds included in the present study). The hepato-specific activities here reported and quantified, clearly support the origin and phenotype of our isolated cells.

Another request submitted by the revisor nr.1 is “a table to indicate the number/type of samples used in all the experiments”. We understood the relevance such information has for the revisor and reader, thus we generated an updated version of Table 1 with all the requested information. We wish such information would be found sufficient to grant publication to our study.

Such a table would clarify as our analyses have been performed on every single donor, with no pool or combined cell analysis. Every single donor or patient has been analyzed separately. Our technology is based on successful human liver cell isolation and functional analysis. We work exclusively with healthy and normal human hepatocytes, isolated from fresh tissue, resected or donated but rejected from solid organ transplantation.

Once again, thanks for helping us in improving our message and discussion.

Reviewer 3 Report

I understood the significance of this study. Authors appropriately added the descriptions about limitations of this study and future perspectives in the Discussion.

Author Response

Thanks for your feedback, we are glad our modifications/adds satisfied the reviewer's requests.